# Examining the viability of five *Salmonella enterica* subsp. *enterica* in thymol at 4°C and 25°C using flow cytometry

Anna Williams[1], Soumana Daddy Gaoh[2], Pierre Alusta[1], Angel Paredes[3], Alena Savenka[3], Dan A. Buzatu[1], Youngbeom Ahn[2]*

1 Division of Systems Biology, National Center for Toxicological Research, U.S. Food and Drug Administration, Jefferson, Arkansas United States of America, 2 Division of Microbiology, National Center for Toxicological Research, U.S. Food and Drug Administration, Jefferson, Arkansas, United States of America, 3 Office of Scientific Coordination, National Center for Toxicological Research, U.S. Food and Drug Administration, Jefferson, Arkansas, United States of America

* young.ahn@fda.hhs.gov

## Abstract

*Salmonella* spp., a major cause of foodborne illness, requires effective control strategies to improve food safety. Thymol, an antibacterial agent derived from natural essential oils, has been assessed for use as an antimicrobial agent and preservative in the food industry, due to its safety and low cost. This study used flow cytometry and Tryptic Soy Agar (TSA) plate counts, to assess the viability of five serotypes of *Salmonella enterica* subsp. *enterica* over 56 days in thymol at 4°C and 25°C, during long-term storage in distilled water. The minimum inhibitory concentration (MIC) of thymol against *Salmonella* serotypes was found to be 256 µg/mL at both 25°C and 4°C for all serotypes at the initial time point (day 0) and after 154 days of incubation in water. Flow cytometry successfully counted viable cells in the control group, which contained 2% ethanol and 128 µg/mL thymol. However, plate count numbers completely declined after day 7 at both 25°C and 4°C for all thymol concentrations. After exposure to sub-MIC levels and subsequent spiking with 256 µg/mL at 25°C and 4°C, neither flow cytometry nor plate counts detected viable cells. These findings emphasize the importance of advanced techniques such as flow cytometry for the detection of microorganisms and demonstrate thymol's potential as an environmentally friendly solution in food safety strategies to reduce *Salmonella* contamination in water sources over extended periods.

## Introduction

The persistence of foodborne pathogens in water systems poses significant challenges to public health and food safety. Among these pathogens, *Salmonella enterica*, a major cause of foodborne illnesses worldwide, has demonstrated the ability to survive under diverse environmental conditions. In the European Union (EU), the

**Data availability statement:** All relevant data are within the manuscript and its Supporting information files.

**Funding:** The authors declare financial support was received for the research, authorship, and/or publication of this article. This research was funded by the U.S. Food and Drug Administration.

**Competing interests:** The authors declare that the research was conducted in the absence of any commercial or financial relationships that could be construed as a potential conflict of interest.

predominant *Salmonella* serotype isolated from food was *S.* Infantis, followed by *S.* Typhimurium, and *S.* Enteritidis [1]. In the United States, *S.* Enteritidis, *S.* Typhimurium, and *S.* Newport represent the most frequently isolated foodborne *Salmonella* serotypes [2]. Controlling the survival and proliferation of *Salmonella* spp. in water systems is essential for mitigating the risk of contamination and subsequent infection [2]. These bacteria can persist in various environmental reservoirs, including natural water systems, drinking water, and irrigation sources, posing a persistent threat to public health and agricultural safety. The ability of *Salmonella* spp. to survive in water systems stems from its physiological adaptability, which allows it to enter a viable but non-culturable (VBNC) state under suboptimal conditions [3]. This state enables bacteria to withstand environmental stresses such as nutrient limitation, temperature fluctuations, and antimicrobial agents, while maintaining the potential to resuscitate and cause infection under favorable conditions [4]. Consequently, even low levels of contamination in water can lead to outbreaks of salmonellosis, especially when the water is used for drinking, food processing, or irrigation of fresh produce. In addition, *Salmonella* spp. can form biofilms on water surfaces, enhancing their resistance to disinfection processes and allowing long-term survival in both treated and untreated water systems [5]. Biofilms not only protect bacteria from environmental stressors but also promote their spread through water distribution networks, increasing the risk of widespread contamination. Additionally, the problem of antibiotic resistance resulting from the overuse of antibiotics is exacerbating bacterial contamination issues.

Addressing these challenges requires innovative approaches to water treatment and management. Disinfectant treatments for life-threatening pathogens are a critical element of entire biosecurity programs [6]. Natural antimicrobial agents, such as thymol (2-isopropyl-5-methylphenol) have gained attention for their potential to inhibit bacterial growth without relying on synthetic chemicals [7]. Thymol, a phenolic compound found in thyme essential oil, exhibits strong antimicrobial properties and serve as a promising alternative for pathogen control in water. Thymol is also included in various personal care products for its antimicrobial and aromatic qualities. This includes hand sanitizers, soaps, acne treatments, and some topical creams and ointments for skin infections. Thymol offers environmentally friendly and sustainable solutions for controlling *Actinobacillus pleuropneumoniae* [8] and *Staphylococcus aureus* [9]. Thymol is also reported to have anti-inflammatory, anti-tumor, and fungicide properties [10–14]. Additionally, thymol has extensively shown immunostimulant activities toward drug-resistant *S.* Enteritidis in chickens, improving weight gain, feed conversion, and survival rates while enhancing protective cytokines and beneficial gut bacteria [15]. Investigating the efficacy of such agents under various environmental conditions is crucial for developing effective strategies to minimize the public health risks posed by *Salmonella* spp. in robust biofilms [6,16–18]. Interestingly, thymol was found to exhibit stronger antibiofilm potential than benzalkonium chloride against *Salmonella enterica* serotype Typhimurium DT193 strain at a concentration of 1250 ppm [19]. However, to our knowledge, more research is needed to understand its long-term efficacy against *S. enterica* across varying temperatures and concentrations is crucial for practical applications.

Recently, our lab reported that all 12 serotypes remain viable after 160 days in distilled water, as determined using the RAPID-B TPC assay [20]. Furthermore, flow cytometry allows for the enumeration and detection of actively growing cells, as well as dormant and/or dead cells [21]. In this study, we evaluated the survival of *Salmonella* serotypes in nuclease-free water treated with thymol at various concentrations (0, 128 µg/mL, 256 µg/mL, and 512 µg/mL), stored at 25°C and 4°C, using flow cytometry. The latter was also used to assess the impact of prior exposure to sub-MICs of thymol on the survival rates of *Salmonella* serotypes, making it a useful tool for analyzing these survival patterns.

## Materials and methods

### Bacterial strains, growth conditions and storage conditions

The five *Salmonella enterica* serotypes used in these experiments are listed in Table 1. Initially, each strain was inoculated onto Tryptic Soy Agar (TSA) plates (Becton, Dickinson and Company, Sparks, MD) and incubated at 37°C for 24 h. The inoculation and storage of *Salmonella* serotypes in autoclaved nuclease-free water (Ambion™, Austin, TX) were performed as previously described by Williams et al. [20]. Briefly, the serotypes suspended in sterile screw-cap bottles containing 1000 mL of sterile autoclaved nuclease-free water, and were then adjusted to a density corresponding to an absorbance of 0.09 to 0.1 at a wavelength of 600 nm (approximate cell density = $10^7$ colony-forming units [CFU/mL]) using the Synergy MX spectrophotometer (BioTek Instruments, Inc., Winooski, VT). The inoculated bottles were divided into two 500 mL portions and stored in two different conditions: one bottle was kept in a cabinet under the benchtop, in the dark, at room temperature (25°C), while the other was stored in a fridge at 4°C for 98–211 days [20].

### Effect of thymol on 5 *Salmonella* serotypes

**Determination of minimum inhibitory concentration (MIC) and minimal bactericidal concentration (MBC).** A broth microdilution assay, using all five *Salmonella* serotypes, was performed to determine the minimal inhibitory concentration (MIC), defined as the lowest concentration of an antimicrobial agent that prevents visible growth of a microorganism after a specified incubation period [22]. Thymol was serially diluted with 20% ethanol (final concentration: 2% ethanol) to final concentrations of 8, 16, 32, 64, 128, 256 or 512 µg/mL in each well. Colonies of five *Salmonella* serotypes were cultivated on TSA at 37°C for 24 h, then resuspended and washed with autoclaved distilled water. Subsequently, they were transferred into 10 mL of autoclaved distilled water, achieving a final inoculum of approximately $10^7$ CFU/mL (optical density, $OD_{600}$ = 0.09 to 0.1). The suspension was then diluted to one-tenth of its original concentration using autoclaved distilled water. In each well of a 96-well plate, 20 µL of the suspended cells (resulting in a final inoculum dose of approximately $10^5$ CFU/mL) and thymol stock solutions were added, along with 160 µL of Mueller-Hinton broth (MHB) media. The control wells contained the medium with the microorganism and 2% ethanol (Decon Labs, Inc., King of Prussia, PA) as the positive control, while the negative control wells included only the medium with 2% ethanol. To eliminate background noise, the optical density (OD) reading for each thymol concentration and the negative

**Table 1. *Salmonella* serotype strains and source.**

| Serotype | Designation and Source | Storage (days) |
|---|---|---|
| *Salmonella enterica* subsp. *enterica serotype* Infantis | *ATCC 706717 | 211 |
| *Salmonella enterica* subsp. *enterica serotype* Javiana | **SAFE DESIGNATON 97 | 181 |
| *Salmonella enterica* subsp. *enterica serotype* Newport | SAFE DESIGNATION 1 | 190 |
| *Salmonella enterica* subsp. *enterica serotype* Enteritidis | SAFE DESIGNATION 81 | 120 |
| *Salmonella enterica* subsp. *enterica serotype* Typhimurium | SAFE DESIGNATION 4 | 98 |

*ATCC = American Type Culture Collection.

**SAFE = Systems and Assays for Food Examination Program.

control was subtracted from the OD reading of wells inoculated with bacteria. All bacterial and control experiments were performed in triplicate. After 41–154 days, aliquots of cell cultures were adjusted to approximately $10^6$ CFU/mL, and then thymol was applied to cells as described above.

To determine the minimal bactericidal concentration (MBC), A 10 µL volume of each *Salmonella* serotype suspension, from the 96-well plates was placed onto TSA plates to recover the bacteria and incubated at 37°C for 48 h. The MBC is defined as the lowest concentration of an antimicrobial agent that inhibits the growth of an organism when subcultured onto antibiotic-free media (Andrews, 2001). Consequently, the thymol concentration at which with no cell growth occurred on thymol- free TSA plates was determined to be the MBC.

**Release of cellular materials outside the cell.** To assess the release of intracellular components, including nucleic acids, metabolites, and proteins, the absorbance of the supernatant was evaluated and reported as concentrations of nucleic acids and proteins (Chauhan and Kang, 2014). *S.* Typhimurium and Enteritidis cells were inoculated and adjusted to approximately $10^5$ CFU/mL with 10 mL, 1/10-diluted TSB into 50 mL conical centrifuge tubes. Thymol was then applied to the cells to achieve final concentrations of 0 (2% ethanol), 32, or 256 µg/mL (dissolved in 2% ethanol). The control tubes contained only the medium with the microorganism and 2% ethanol, without thymol. The cultures were incubated at 37°C for 7 h, and samples were taken at the following intervals: 0, 30, 60, 90, and 360 min (6 h). At each interval, 1 mL of each Salmonella serotype culture was centrifuged in an Eppendorf 5425 microcentrifuge for 1 minute at 3,500 rpm. Subsequently, 1 µL of the supernatant was analyzed at 260 nm and 280 nm using a NanoDrop™ One/One<sup>C</sup> Microvolume UV-Vis Spectrophotometer (ThermoFisher Scientific).

**Scanning electron microscopy (SEM).** The effect of thymol on the morphology of *S.* Typhimurium and *S.* Enteritidis serotypes was examined using a Zeiss Merlin Gemini2 Field Emission Scanning Electron Microscope (SEM – Carl Zeiss Company, LLC, White Plains, NY). The day 0 culture, which was not exposed to thymol, served as the control for the SEM images [20]. After 98 days exposure for *S.* Typhimurium and 120 days for the *S.* Enteritidis, both serotypes were incubated at 25°C and at 4°C. Then, 900 µL aliquots were transferred into two 2 mL sterile microcentrifuge tubes (Costar, Corning Inc., Cambridge, MA). Control samples received 100 µL of 2% ethanol only, while thymol solutions (100 µL, final concentrations of 32 µg/mL and 256 µg/mL) were added to the *S.* Typhimurium and *S.* Enteritidis serotypes. Scanning electron microscopy was performed as previously described where bacterial pellets were dehydrated through an increasing ethanol concentration series and dried in hexamethyldisilazane (HMDS) to preserve fine structural detail. Samples were then coated with gold-palladium using a Denton Desk V sputter coater (Denton Vacuum, Moorestown, NJ.) [20].

## Viability in thymol-treated water of long-term survival of *Salmonella enterica* serotype

**Direct viable plate count method.** Among the five *Salmonella* serotypes at both 25°C and 4°C, *S.* Typhimurium (incubated for 63 days) and *S.* Enteritidis (incubated for 41 days, with an initial concentration of approximately $10^7$ CFU/mL) were selected for cell recovery comparison. The thymol concentrations were adjusted to 0 (2% ethanol), 128 (sub-MIC), 256 (MIC) and 512 µg/mL. These thymol-treated water samples were incubated at both 25°C and 4°C for 56 days. Samples were collected on day 1, 7, 28, and 56. At each time point, the collected samples were serially diluted with sterile water and 10 µL of serial dilutions plated on TSA media to enumerate bacterial cells [20,23,24]. The plates were then incubated at 37°C for 24 h, and the number of colony forming units (CFU) on each plate was counted using the ProtoCOL3 automated plate counter (Synbiosis, Frederick, MD). Plate assays were performed in three to six replicates, and the results were averaged. Images of each dilution on plates were also captured using the ProtoCOL3 and saved for comparison and future reference.

**RAPID-B total plate count (TPC).** A flow cytometer (Apogee Model A50, Apogee Flow Systems, Catalonia, Spain), in combination with a total plate count assay (TPC) comprising two DNA dyes – propidium iodide (Sigma-Aldrich, Steinheim, Germany) and thiazole orange (Sigma-Aldrich Steinheim, Germany) – was used to count viable, culturable

cells and non-viable, non-culturable cells in real time [20]. To determine the number of viable cells from *S.* Typhimurium and *S.* Enteritidis serotypes stored in thymol-treated water, the cells described above were treated with TPC reagent and analyzed using flow cytometry. At each time point for each serotype, 667 µL of the sample was taken, and 333 µL of the TPC reagent added before sample analysis [25]. Each sample was analyzed in triplicate and the event counts per 100 µL were recorded. The flow cytometric data were converted to cells per mL and compared to colony forming units per mL (CFU/mL) obtained from plate counts.

**Salmonella serotype response to thymol tolerance.** To investigate *Salmonella* thymol tolerance, we intentionally spiked approximately $10^7$ CFU/mL *of S.* Typhimurium and *S.* Enteritidis incubated respectively after 70 and 48 days in water at 4 and 25°C with their thymol sub-MICs. Briefly, both strains were spiked with thymol to achieve final concentrations of 0 (2% ethanol), 32, 64 and 128 µg/mL of thymol, then incubated for 36 days at 4 and 25°C. The survival of these serotypes in thymol-treated water was evaluated after 1, 23, and 36 days. At each time point, 10 µL of each treated serotype serial dilution was plated on TSA and incubated at 37°C for 48 h to recover the bacteria. Additionally, these dilutions were analyzed by flow cytometry as described previously.

Finaly, on day 37, another dose of thymol was introduced into both strains to achieve a final thymol concentration of 256 µg/mL (MIC). Subsequently, the direct viable plate count method and the RAPID-B TPC assay were conducted on days 37 and 42, following the same procedures outlined previously.

## Statistical analysis

Graphs represent averages of triplicate samples. Significant differences between the detection of intracellular materials (nucleic acids and protein materials), number of colony forming units and TPC assay using RAPID-B flow cytometry for *S.* Typhimurium and *S.* Enteritidis were determined using SigmaPlot vs. 14.1 software (GraphPad Software, Inc. USA).

## Results

### Viability assessment of *Salmonella enterica* subsp. enterica serotypes

**Susceptibility of *Salmonella* serotype to thymol.** The susceptibility of freshly cultured *Salmonella* serotypes to thymol was compared to that of the same serotypes incubated in distilled water for 41–154 days (Table 2, S1 Fig). The susceptible concentration of thymol (on day 0) was 256 µg/mL. Among the five *Salmonella* serotypes stored in water at 25°C and 4°C, *S.* Typhimurium (stored for 63 days) and *S.* Enteritidis (stored for 41 days) reached a concentration of $10^7$ CFU/mL, which was sufficient for susceptibility testing. Both serotypes exhibited susceptibility to thymol at a concentration of 256 µg/mL, regardless of storage at 25°C and 4°C. However, *S.* Javiana and *S.* Infantis, stored for 124 and 154 days

**Table 2. Concentrations of thymol that cause *Salmonella* serotype susceptibility (inoculation cell density: $10^5$ CFU/mL) incubated at 37°C for 24 h.**

| Serotype | Initial day (0 day) in distilled water at 25°C (µg/mL) | | After storage in distilled water | | | | Store date (days) |
|---|---|---|---|---|---|---|---|
| | | | 25°C (µg/mL) | | 4°C (µg/mL) | | |
| | MIC | MBC | MIC | MBC | MIC | MBC | |
| Javiana | 256 | 512 | 256 | 512 | 128* | 512 | 124 |
| Typhimurium | 256 | 512 | 256 | 512 | 256 | 512 | 63 |
| Enteritidis | 256 | 512 | 256 | 512 | 256 | 512 | 41 |
| Newport | 256 | 512 | 256** | 512 | 256* | 512 | 133 |
| Infantis | 256 | 512 | 256 | 512 | 256** | 512 | 154 |

*final cell number is $10^3$ CFU/mL.

**final cell number is $10^4$ CFU/mL.

of incubation at 4°C, respectively, reached concentrations of $10^4$–$10^5$ CFU/mL, which was insufficient for susceptibility testing. Similarly, *S.* Newport reached a concentration of $10^4$–$10^5$ CFU/mL after 133 days of incubation at both 25°C and 4°C. Despite the lower cell density, *S.* Infantis and *S.* Newport retained the same thymol susceptible concentration (256 µg/mL) at both temperatures. In contrast, *S.* Javiana (with a final cell density of approximately $10^3$ CFU/mL) at 4°C exhibited susceptibility to a lower concentration of thymol (128 µg/mL) compared to the initial inoculation.

The bactericidal activity of thymol (MBC) for all five *Salmonella* serotypes at both 25°C and 4°C was confirmed at 512 µg/mL, as shown in Table 2.

## Release of cellular materials outside the cell

*S.* Typhimurium and *S.* Enteritidis were suspended in solutions containing 0 (no thymol, 2% ethanol), 32, or 256 µg/mL of thymol. Fig 1 illustrates the release of cellular materials from cells treated with MIC thymol (256 µg/mL) compared to

**Fig 1. Intracellular materials were released from *S.* Typhimurium (A and B) and *S.* Enteritidis (C and D).** Thymol concentrations of 0 (2% ethanol), 32 or 256 µg/mL (dissolved in 2% ethanol) were added to *Salmonella* serotype and the supernatant was measured at 260 nm (A and C) and 280 nm (B and D) in NanoDrop™ One/One© Microvolume UV-Vis Spectrophotometer. The graphs indicators were significantly lower in the 0 (2% ethanol), and 32 µg/mL than in the 256 µg/mL based on the result of the Mann-Whitney Rank Sum test ($p < 0.05$).

untreated control cells or those exposed to sub-MIC thymol (32 µg/mL). MIC thymol solutions caused a sharp increase in the mean nucleic acid concentration detected at 260 nm for *S.* Typhimurium (Fig 1A) and *S.* Enteritidis (Fig 1C) rising to 107.1 ± 0.6 ng/µL and 84.3 ± 10.0 ng/µL, respectively, after 10 min of incubation. These values further increased to 186.3 ± 7.5 ng/µL and 153.1 ± 6.5 ng/µL at 360 min (6 h), compared to the control samples incubated in water without thymol. In contrast, sub-MIC thymol (32 µg/mL) resulted in lower nucleic acid concentrations for *S.* Typhimurium and *S.* Enteritidis (44.1 ± 9.7 ng/µL and 31.8 ± 11.0 ng/µL, respectively). These values remained consistently higher than those of untreated samples, which had nucleic acid concentrations of 29.2 ± 12.3 ng/µL and 16.2 ± 13.6 ng/µL, respectively.

Protein concentrations detected at 280 nm for *S.* Typhimurium (Fig 1B) and *S.* Enteritidis (Fig 1D) followed a pattern similar to that observed for nucleic acid concentrations detected at 260 nm. The average protein concentrations in cells treated with MIC thymol solutions were 4.3 ± 0.7 mg/mL for *S.* Typhimurium and 4.5 ± 0.9 mg/mL for *S.* Enteritidis. However, sub-MIC thymol solutions resulted in low protein concentrations (0.9 ± 0.3 mg/mL and 1.5 ± 0.7 mg/mL), which were consistently higher than those detected in untreated control samples (0.2 ± 0.3 mg/mL and 0.7 ± 0.7 mg/mL) throughout the 360-minute incubation period.

### Scanning electron microscopy

Morphological changes from treatment with 256 µg/mL thymol (MIC) for 24 h and 48h were observed in *S.* Typhimurium by SEM (Fig 2B and C). The control without thymol (2% ethanol) exhibited rod shapes and smooth surfaces (Fig 2A). In contrast, the 256 µg/mL thymol-treated cells were irregularly sized in presence of debris and pores on the surface, possibly by dysfunction of cellular membrane (Fig 2B and C). The release of intracellular materials (*i.e.*, nucleoid, ribosomes, plasmids, RNA molecules and larger proteins) – clearly visible in Fig 2B and C – is due to the disruption of bacterial cell wall facilitated by the presence of thymol. The solubility of the latter in organic solvents, such as alcohols, ethers and

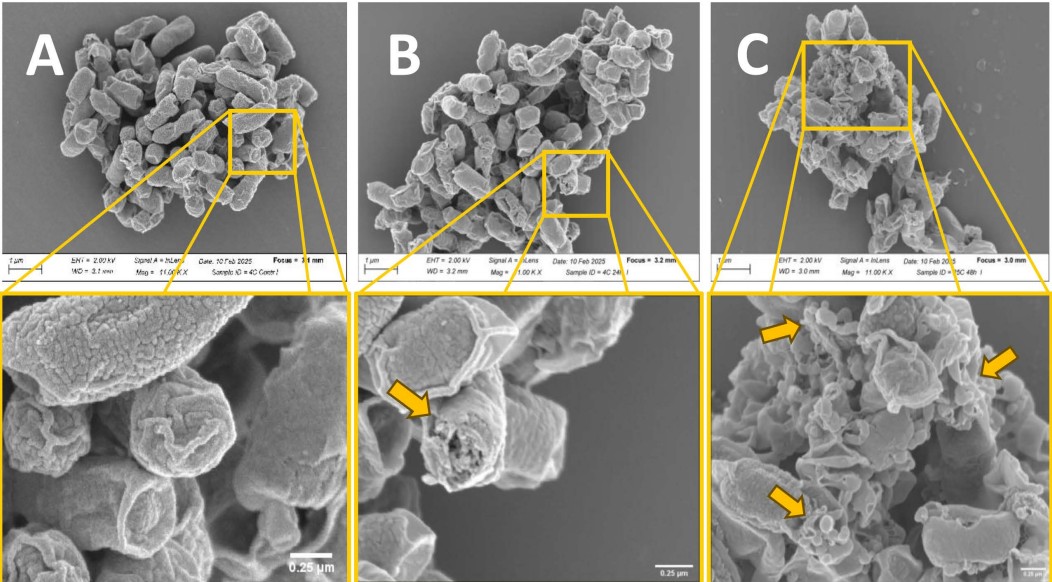

**Fig 2. Morphology change of *S.* Typhimurium in solution without thymol (2% ethanol) (A), with 256 µg/mL thymol incubated for 24h (B), and with 256 µg/mL thymol incubated for 48 h (C).** Scales in all micrographs: 1 µm. Magnification: 11,000×. The control consisted of bacterial cells at initial growth at 25°C. Note the release of intracellular materials (*i.e.*, nucleoid, ribosomes, plasmids, RNA molecules and larger proteins, see yellow arrows) in B and C due to disruption of bacterial cell wall facilitated by the presence of thymol. Although cell shriveling is attributed to the loss of water during sample preparation for SEM microscopy, it is not believed to be a contributing factor to bacterial cell wall perforation.

lipids, which enables the dissolution of lipids, which accounts for the observed cell wall perforation as seen in Fig 2B and C. Although ostensible cell shriveling is attributed to the loss of water during sample preparation for SEM microscopy (Fig 2A), it is not believed to be a contributing factor to bacterial cell wall perforation as seen in Fig 2B and C.

### Survival in various concentrations of thymol-treated water; comparison of the flow cytometric assay to traditional agar plate counts

Both flow cytometry and plate counts successfully detected viable cells initially (S2 and S3 Figs). However, since thymol inhibits bacterial recovery, the detection of surviving cells after one day varied depending on the thymol concentration. In the control group (2% ethanol), *S.* Typhimurium and *S.* Enteritidis exhibited similar cell counts at 25°C (1.70 to 1.87 × 10⁶ CFU/mL, respectively) and at 4°C (1.33 to 4.33 × 10⁵ CFU/mL, respectively) up to day 1. Subsequently, the survival of *S.* Typhimurium declined, with cell counts decreasing to $2.34 \times 10^3$ CFU/mL at 25°C (Fig 3A) and $6.67 \times 10^2$ CFU/mL at 4°C (Fig 3B) by day 28. By day 56, no growth was detected at either temperature. The survival pattern of *S.* Enteritidis at 25°C

**Fig 3. Comparative recovery of *S.* Typhimurium and *S.* Enteritidis on TSA plates at 25°C (A and C) and 4°C (B and D).** Thymol concentrations were adjusted to 0 (2% ethanol), 128 (sub-MIC), 256 (MIC) and 512 µg/mL.

(Fig 3C) was similar to that of *S.* Typhimurium with cell counts decreasing to $1.67 \times 10^3$ CFU/mL at day 28. However, at 4°C (Fig 3D), *S.* Enteritidis exhibited a more rapid decline, with cell counts dropping to $2.33 \times 10^4$ CFU/mL on day 7, and no detectable growth by day 28. Exposure to thymol at 128 µg/mL significantly reduced the recovery of *S.* Typhimurium, which decreased to $2.31 \times 10^4$ CFU/mL at 25°C by day 7. However, no viable *S.* Typhimurium cells were detected on TSA at 4°C, nor were *S.* Enteritidis cells detected at either at 25°C and 4°C after day 7. When treated with thymol at 256 and 512 µg/mL, *S.* Typhimurium and *S.* Enteritidis at both 25°C and 4°C exhibited no growth on TSA even after day 1.

A comparison of the sensitivity of flow cytometry in detecting *S.* Typhimurium and *S.* Enteritidis at thymol concentrations ranging from 0 to 512 µg/mL, stored at 25ºC and 4ºC for 56 days, is presented in Fig 4. At 0 µg/mL thymol (2% ethanol), *S.* Typhimurium (Fig 4A and 4C) and *S.* Enteritidis (Fig 4B and 4D) were maintained at 25°C (1.05 to $3.11 \times 10^6$ cells/mL, respectively) and at 4°C (1.25 to $2.71 \times 10^6$ cells/mL, respectively) until day 1. After that, *S.* Typhimurium and *S.* Enteritidis cell counts slightly decreased to $3.16–7.51 \times 10^5$ cells/mL at 25°C and $5.31–32.0 \times 10^5$ cells/mL at 4°C by day 56. Thymol at 128 µg/mL significantly reduced the mean cell count of *S.* Typhimurium and *S.* Enteritidis, decreasing



**Fig 4. Sensitivity of TPC assay using RAPID-B flow cytometry in detecting *S.* Typhimurium and *S.* Enteritidis at 25°C (A and C) and 4°C (B and D).** Thymol concentrations were adjusted to 0 (2% ethanol), 128 (sub-MIC), 256 (MIC) and 512 µg/mL. The graphs indicators were significantly lower in the 0 (2% ethanol), than in the 128 (sub-MIC) µg/mL ($p < 0.05$).

to $4.6 \times 10^3$–$5.2 \times 10^4$ cells/mL at 25°C and $1.5 \times 10^4$–$2.99 \times 10^7$ cells/mL, respectively, at 4°C after 7 days of incubation. Thereafter, bacterial survival further declined, reaching $3.67 \times 10^3$–$5.0 \times 10^3$ cells/mL, respectively at 25°C and 11.7 to $70.7 \times 10^3$ cells/mL at 4°C by day 56. Both 256 and 512 μg/mL thymol-treated water caused a sharp reduction in the mean cell count of S. Typhimurium and S. Enteritidis at 25°C (from $4.15 \times 10^3$ to $6.2 \times 10^5$ cells/mL, respectively) and at 4°C ($8.03 \times 10^2$ to $1.03 \times 10^5$ cells/mL, respectively) after day 1. After day 7, S. Typhimurium and S. Enteritidis were not detect by flow cytometry at both 25°C and 4°C.

### *Salmonella* serotype response to thymol tolerance

A thymol tolerance experiment was conducted to determine whether S. Typhimurium and S. Enteritidis exhibited increased resistance to thymol in the presence of sub-MIC thymol concentrations. Water-adapted S. Typhimurium and S. Enteritidis were suspended in solutions containing 0 (2% ethanol), 32, 64, and 128 μg/mL of thymol. On day 1, S. Typhimurium (Fig 5A and 5B) and S. Enteritidis (Fig 5C and 5D) showed cell counts at 25°C ranging from $5.34 \times 10^5$ to $17.0 \times 10^5$ CFU/

**Fig 5. Effects of MICs on the growth of S. Typhimurium (A and B) and S. Enteritidis (C and D) on TSA plates prior to exposure to sub-inhibitory concentrations of thymol (0 (2% ethanol), 32, 64 and 128 μg/mL).** Arrow (at 36 days) indicate when 256 μg/mL of thymol were added.

mL and at 4°C ranging from $1.33 \times 10^5$ to $8.80 \times 10^5$ CFU/mL. Following this, in the control (2% ethanol), 32, and 64 µg/mL of thymol treatments, *S.* Typhimurium and *S.* Enteritidis showed a slight decrease in cell counts at 25°C, ranging from $1.50 \times 10^3$ to $30 \times 10^3$ CFU/mL) until day 23. However, viable *S.* Enteritidis cells were not detected on TSA at 4°C after day 23. The control (2% ethanol) and 32 µg/mL thymol treatments showed further declines in the recovery of *S.* Typhimurium and *S.* Enteritidis at 25°C (1.1 to $4.3 \times 10^2$ CFU/mL) and *S. Typhimurium* at 4°C (1.0 to $2.7 \times 10^2$ CFU/mL) until day 36. At 128 µg/mL thymol, *S.* Typhimurium and *S.* Enteritidis were not detected at either 25°C and 4°C after 23 days. When 256 µg/mL thymol was added to the treated water, *S.* Typhimurium and *S.* Enteritidis were not detected at either 25°C and 4°C on day 37.

The detection of *S.* Typhimurium and *S.* Enteritidis using flow cytometry at 25ºC and 4ºC over 36 days is presented in [Fig 6]. On day 1, S. Typhimurium ([Fig 6A] and [6B]) and *S.* Enteritidis ([Fig 6C] and [6D]) showed cell counts of $2.2 \times 10^5$ to $10.5 \times 10^5$ cells/mL at 25°C and $12.5 \times 10^5$ to $45.7 \times 10^5$ cells/mL at 4°C. Subsequently, in the control 0 (2% ethanol), 32 and 64 µg/mL thymol treatments, *S.* Typhimurium and *S.* Enteritidis remained at 25°C ($0.19 \times 10^5$ to $18.4 \times 10^5$ cells/mL) and 4°C ($9.72 \times 10^5$ to $38.1 \times 10^5$ cells/mL) until day 36. Treatment with 128 µg/mL thymol sharply reduced the cell counts



**Fig 6. The detection of *S.* Typhimurium (A and B) and *S.* Enteritidis (C and D) using RAPID-B TPC flow cytometry prior to exposure to sub-inhibitory concentrations of thymol (0 (2% ethanol), 32, 64 and 128 µg/mL).** Arrow (at 36 days) indicate when 256 µg/mL of thymol were added.

of *S.* Typhimurium and *S.* Enteritidis to $7.0 \times 10^3$ to $17.3 \times 10^3$ cells/mL at 25°C and $79.3 \times 10^3$ to $6.05 \times 10^5$ cells/mL at 4°C by day 36. After 37 days of treatment with 256 µg/mL thymol, *S.* Typhimurium and *S.* Enteritidis were not detected by flow cytometry at either 25°C and 4°C.

## Discussion

*Salmonella* is a foodborne pathogen that can survive in water for prolonged periods [4,20,26]. In this study, five *Salmonella* serotypes stored in water at both 25°C and 4°C for 41–154 days exhibited susceptibility to certain concentrations of thymol. A meta-analysis on thymol MIC, based on MIC values found in the scientific literature, showed median values of 317 µg/mL for molds, 513 µg/mL for yeasts, 400 µg/mL for Gram-negative microorganisms, and 317 µg/mL for Gram-positive microorganisms [27]. *Salmonella* species exhibited high variability in MIC values, ranging from 0.5 to 400 µg/mL. A previous study reported an MIC of 750 µg/mL for thymol (dissolved in 5% dimethyl sulfoxide) against *Salmonella enterica* serovar Typhimurium ATCC 14028 [28]. In the present study, similar values were observed, with the MIC and MBC for thymol against *S.* Typhimurium and *S.* Enteritidis being 256 µg/mL and 512 µg/mL, respectively. Considering that thymol is used in various commercial products at concentrations up to 0.064% (640 µg/mL; *e.g.,* Listerine mouthwash), this concentration might be sufficient to kill *Salmonella* [29]. These results align with previous studies demonstrating MIC (93.4 µg/mL) and MBC (373.6 µg/mL) values for *Streptococcus mutans* and *S. sanguinis* [30]. Additionally, carvacrol and thymol at 200 µg/mL were shown to inhibit the growth of *E. coli* [31]. Similar results were obtained for *Candida albicans* and *Candida krusei*, with MIC values of 39 and 78 µg/mL, respectively [32]. Our data underscore the importance of proper thymol dilution levels in products to ensure effectiveness as disinfectants.

Thymol acts on the cell membrane, causing leakage of cell cytoplasmic materials, which ultimately leads to cell death [31,33]. Xu et al. [31] confirmed that the antibacterial effect is related to the induction of permeabilization and depolarization of the cytoplasmic membrane. Furthermore, Trombetta et al. [34] evaluated the ability of thymol and other monoterpenes to damage bacterial lipid membranes, in order to better understand thymol's mechanisms of action. In this study, the observed release of nucleic acids and proteins, along with the SEM results, collectively indicate that MIC concentrations of thymol can induce severe membrane damage in *Salmonella* serotypes. In contrast, sub-MIC thymol (32 µg/mL) and the control (2% ethanol) treatments – which lacked thymol – resulted in a low release of nucleic acids and proteins, as well as a smooth cell surface. Thymol disrupts bacterial cell membranes primarily due to its lipophilic nature. Its structure allows it to insert itself into the lipid bilayer of the bacterial membrane. Thymol's presence at the bacterial cell membrane disturbs the tightly packed arrangement of lipids, which increases the membrane's fluidity and permeability [35]. This instability leads to the leakage of essential intracellular materials (*i.e.*, nucleoid, ribosomes, plasmids, RNA molecules and larger proteins) as shown in Fig 2C. At higher thymol concentrations, the membrane disruption becomes so severe that it causes visible structural damage to the cell surface [36], as seen in SEM micrographs. This can lead to the observed cell lysis, or bursting. The results we obtained are consistent with previous studies that reported the release of intracellular components and increased extracellular potassium ion concentrations [28]. Furthermore, visualization of thymol-induced morphological alterations in the cell membrane was performed using SEM [28,34] and transmission electron microscopy (TEM) [8]. The alteration of the cell membrane's morphology and the release of intracellular components caused by thymol may have affected not only bacterial sensitivity to thymol but also bacterial survival, possibly by disrupting the cell walls and membranes.

The RAPID-B TPC flow cytometric assay can distinguish between live and dead cells [20,25]. Most recently, the RAPID-B TPC flow cytometric assay, when compared to traditional growth plates, demonstrated that *Salmonella* serotypes are capable of surviving in distilled water for over 160 days at both 25°C and 4°C [20]. In this study, *Salmonella* serotypes treated with thymol failed to grow on TSA plates, but were detected using the RAPID-B TPC assay. It is not surprising that in the control (2% ethanol) and sub-MIC (128 µg/mL) conditions, *S.* Typhimurium and *S.* Enteritidis did not show growth at either temperature by day 56, whereas all concentrations remained detectable using flow cytometry at both temperatures.

We observed that bacterial cells were stressed in both the control (2% ethanol) and thymol-treated water, and traditional methods for bacterial detection and characterization were limited. Nonetheless, the control (2% ethanol) is unlikely to be effective as a disinfectant or antiseptic. The present study confirms that RAPID-B flow cytometry enables the detection and quantification of *Salmonella* serotypes in thymol-treated water, providing a more accurate assessment of bacterial viability.

Thymol is a safe and inexpensive compound that has been evaluated for use as an antimicrobial agent and preservative in the food industry. In commercial products, the use of disinfectants or antiseptics requires consideration of bacteriostatic parameters; therefore, the applied doses are higher than the MIC. Antiseptics or disinfectants at sub-MIC levels are unable to inhibit bacterial proliferation. However, they can cause morphological and physiological changes in *Pseudomonas aeruginosa* [37] and *Staphylococcus aureus* [9]. Antiseptic resistance genes (*qacAB, smr, qacG, qacH, qacJ*), which encode multidrug efflux pumps carried by plasmids, have been identified in *Staphylococcus* spp. and *Enterococcus* spp. [38]. Previous studies indicate that when efflux pumps are activated by tetracycline, the addition of sub-MIC of thymol enhances the antibiotics effectiveness [39]. Since thymol is known to create pores on the surface of bacteria, it is also recognized for directly inhibiting efflux pumps activity [40–42]. The results of the present study showed that, as expected, when *S.* Typhimurium and *S.* Enteritidis were exposed to sub-MIC levels of thymol for a period of time and then treated with MIC levels of thymol, they did not survive on TSA plates and were not detected by flow cytometry. These data suggest that thymol may be useful for the treatment of *Salmonella* infections without leading to the selection and evolution of resistant serotypes.

## Conclusion

This study demonstrated that thymol kills *Salmonella* serotypes predominantly by disrupting microbial cell membranes, resulting in the release of intracellular materials. RAPID-B flow cytometric analysis can be successfully employed to count viable cells in thymol-treated samples, whereas plate count numbers decreased, limiting detection. At 128 µg/mL thymol, *Salmonella* serotypes remained detectable at 25°C and 4°C until day 56 using flow cytometry, but viable cells could not be counted on plates. To assess the impact of prior exposure to sub-inhibitory concentrations of thymol, *Salmonella* serotypes did not exhibit increased survival rates when subsequently treated with 256 µg/mL thymol. Therefore, this study is in favor of using advanced techniques such as flow cytometry and highlights thymol's potential as a safe and inexpensive antimicrobial agent. In food safety, thymol's Generally Recognized as Safe (GRAS) status by the FDA allows for its use as a preservative in certain foods. Its efficacy and low toxicity is conducive to its use as a preservative and disinfectant. As a preservative, thymol can be incorporated into food products or packaging to inhibit the growth of common foodborne pathogens such as *Salmonella* spp., thus extending shelf life and enhancing food quality assurance. For work surfaces, thymol is a key active ingredient in botanical disinfectants and sanitizers, providing an alternative to traditional chemical agents. Thymol-based products are effective for decontaminating food-contact and non-food-contact surfaces, reducing the risk of cross-contamination in food processing facilities, kitchens, and other environments.

## Supporting information

**S1 Fig. A schematic experimental timeline.** This figure visually represents the sequence of experimental steps involving *Salmonella enterica* serotype strains subjected to different treatments over varying durations. The flow chart consists of arrows, each corresponding to a specific phase of the experiment. Phase 1: Initial storage and incubation. Phase 2: Viability in thymol-treated water. Phase 3: Thymol tolerance testing.
(DOCX)

**S2 Fig. Screenshots of FL1 vs. FL3 fluorescence plot.** (A) blank sample containing phosphate buffer (PBS) and total plate count (TPC) reagent alone; (B) Day 7 – *S.* Enteritidis with sub-MIC 128 µg/mL thymol; (C) Day 7 – *S.* Enteritidis with sub-MIC 256 µg/mL thymol; and (D) Day 7 – *S.* Enteritidis with sub-MIC 512 µg/mL thymol.
(DOCX)



**S3 Fig. Plate images of *Salmonella* serotypes.** *Salmonella* Typhimurium (A, B, C, and D) and Enteritidis (E, F, G, and H) exposed to 0 μg/mL (2% ethanol, A and E), 128 μg/mL (B and F), 256 μg/mL (C and G), and 512 μg/mL (D and H) of thymol after 7 days on TSA plates at 25°C and 4°C. The left side of each plate was inoculated with cells incubated at 25°C and the right side of each plate inoculated with cells incubated at 4°C. For comparison with various concentrations of thymol, a series of four dilutions was plated (three times) at 25°C and 4°C and imaged using the ProtoCOL3 automated plate counter.
(DOCX)

## Acknowledgments

We thank Drs. Huizhong Chen and Vasily Dobrovolsky for critical review of the manuscript. This manuscript reflects the views of the authors and does not necessarily reflect those of the Food and Drug Administration.

## Author contributions

**Conceptualization:** Pierre Alusta, Youngbeom Ahn.

**Data curation:** Dan A. Buzatu, Alena Savenka, Angel Paredes.

**Formal analysis:** Alena Savenka, Angel Paredes, Soumana Daddy Gaoh, Anna Williams.

**Investigation:** Youngbeom Ahn, Soumana Daddy Gaoh, Anna Williams.

**Methodology:** Dan A. Buzatu.

**Writing – review & editing:** Pierre Alusta.

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
