## [Decision Letter · Decision Letter 0]

25 Jul 2025

PONE-D-25-33553

Examining the viability of five Salmonella enterica subsp. enterica in thymol at 4°C and 25°C using flow cytometry

PLOS ONE

Dear Dr. Ahn,

Thank you for submitting your manuscript to PLOS ONE. After careful consideration, we feel that it has merit but does not fully meet PLOS ONE’s publication criteria as it currently stands. Therefore, we invite you to submit a revised version of the manuscript that addresses the points raised during the review process.

We look forward to receiving your revised manuscript.

Kind regards,

Jorddy Neves Cruz

Academic Editor

PLOS ONE

[The authors declare that the research was conducted in the absence of any commercial or financial relationships that could be construed as a potential conflict of interest.].

 [The authors declare financial support was received for the research, authorship, and/or publication of this article. This research was funded by the U.S. Food and Drug Administration.]. 

6. Please include a copy of Table 1-2 which you refer to in your text on page 9.

Additional Editor Comments (if provided):

Reviewers' comments:

Reviewer's Responses to Questions

**Comments to the Author**

1. Is the manuscript technically sound, and do the data support the conclusions?

Reviewer #1: Yes

Reviewer #2: Yes

2. Has the statistical analysis been performed appropriately and rigorously? 

Reviewer #1: N/A

Reviewer #2: No

3. Have the authors made all data underlying the findings in their manuscript fully available?

Reviewer #1: Yes

Reviewer #2: Yes

4. Is the manuscript presented in an intelligible fashion and written in standard English?

Reviewer #1: Yes

Reviewer #2: Yes

5. Review Comments to the Author

Reviewer #1: Abstract -if have word count available- give an example of types of products thymol is used in if any

Introduction-

-Italicize salmonella names-check this overall in text

-Line 77 include Salmonella in all serotypes line just so clear you mean all salmonella serotypes also our lab sounds more professional than we if talking about previous studies not just current one.

-Is thymol just used in water? Is there more products it is in? Does this currently have a commercial use or is it just studied alternative. This is not clear.

Materials and Methods:

-no comments

Results

-no comments

Discussion + conclusions

-no comments

Figures

-figures need titles and figure legends. Hard to understand currently

Reviewer #2: This study provides valuable insights into the antimicrobial effects of thymol on Salmonella enterica, with potential applications in food and water safety. However, the following revisions are required to strengthen the manuscript:

1. Statistical analysis: The methods section should specify the number of replicates used in the experiments. Statistical significance (e.g., p-values) must be clearly reported to support the conclusions. Provide error bars in the graphs.

2. Data interpretation: The observed reduction in protein materials and increase in nucleic acid content requires further explanation. Please discuss the possible biological mechanisms underlying these changes.

3. Practical implications: The Conclusion section should explicitly address how these findings could be applied in real-world food safety or disinfection protocols.

4. Please highlight the advantages of flow cytometry over viable plate count method.

5. Language editing: There are some grammar errors (for instance line 180). A thorough grammar and style review is recommended to improve readability (e.g., subject-verb agreement, article usage).

6. PLOS authors have the option to publish the peer review history of their article (what does this mean? ). If published, this will include your full peer review and any attached files.

**Do you want your identity to be public for this peer review?** For information about this choice, including consent withdrawal, please see our Privacy Policy .

Reviewer #1: **Yes: ** Megan L. Fay

Reviewer #2: No

---

## [Author Response · Author response to Decision Letter 1]

14 Aug 2025

Reviewer #1:

Abstract -if have word count available- give an example of types of products thymol is used in if any

We appreciate this valuable comment! Unfortunately, we have not word count available in Abstract. We have added the following statement in the Introduction section on lines 66-68, namely. “Thymol is also included in various personal care products for its antimicrobial and aromatic qualities. This includes hand sanitizers, soaps, acne treatments, and some topical creams and ointments for skin infections.”

Introduction-

-Italicize salmonella names-check this overall in text

As recommended, we have italicized Salmonella names. However, serotypes should not be put in italics.

-Line 77 include Salmonella in all serotypes line just so clear you mean all salmonella serotypes also our lab sounds more professional than we if talking about previous studies not just current one.

Thank you for pointing this out! Since we have provided references, we though it to be superfluous to specify which Salmonella serovars we were referring to (see Line 81). We have therefore used “our lab” instead of “we” to maintain a professional tone.

-Is thymol just used in water? Is there more products it is in? Does this currently have a commercial use or is it just studied alternative. This is not clear.

We appreciate this valuable comment! We have added the following statement in the Introduction section on lines 66-68, namely. “Thymol is also included in various personal care products for its antimicrobial and aromatic qualities. This includes hand sanitizers, soaps, acne treatments, and some topical creams and ointments for skin infections.”

Materials and Methods:

-no comments

Results

-no comments

Discussion + conclusions

-no comments

Figures

-figures need titles and figure legends. Hard to understand currently

We added separate captions for each figure in the manuscript.

Reviewer #2: This study provides valuable insights into the antimicrobial effects of thymol on Salmonella enterica, with potential applications in food and water safety. However, the following revisions are required to strengthen the manuscript:

1. Statistical analysis: The methods section should specify the number of replicates used in the experiments. Statistical significance (e.g., p-values) must be clearly reported to support the conclusions. Provide error bars in the graphs.

As recommended, we have added the following statement in the Material and Methods section on lines 195-199, “Statistical analysis Graphs represent averages of triplicate samples. Significant differences between the detection of intracellular materials (nucleic acids and protein materials), number of colony forming units and TPC assay using RAPID-B flow cytometry for S. Typhimurium and S. Enteritidis were determined using SigmaPlot vs. 14.1 software (GraphPad Software, Inc. USA).” Furthermore, we added in the Fig. 1 legend the following description: “The graphs indicators were significantly lower in the 0 (2% ethanol), and 32 µg/mL than in the 256 µg/mL based on the result of the Mann-Whitney Rank Sum test (p < 0.05).” and Fig. 4 legend the following description: “The graphs indicators were significantly lower in the 0 (2% ethanol), than in the 128 (sub-MIC) µg/mL (p < 0.05).

2. Data interpretation: The observed reduction in protein materials and increase in nucleic acid content requires further explanation. Please discuss the possible biological mechanisms underlying these changes.

As recommended, we have added the following statement in the Discussion section on lines 334-341, “Thymol disrupts bacterial cell membranes primarily due to its lipophilic nature. Its structure allows it to insert itself into the lipid bilayer of the bacterial membrane. Thymol's presence at the bacterial cell membrane disturbs the tightly packed arrangement of lipids, which increases the membrane's fluidity and permeability [35]. This instability leads to the leakage of essential intracellular materials (i.e., nucleoid, ribosomes, plasmids, RNA molecules and larger proteins) as shown in Fig. 2C. At higher thymol concentrations, the membrane disruption becomes so severe that it causes visible structural damage to the cell surface [36], as seen in SEM micrographs. This can lead to the observed cell lysis, or bursting.” We believe this information sufficiently addresses the concern raised by the reviewer.

3. Practical implications: The Conclusion section should explicitly address how these findings could be applied in real-world food safety or disinfection protocols.

For clarification, we have improved the following sentence in the Conclusion on lines 383-392. “Therefore, this study is in favor of using advanced techniques such as flow cytometry and highlights thymol's potential as a safe and inexpensive antimicrobial agent. In food safety, thymol's Generally Recognized as Safe (GRAS) status by the FDA allows for its use as a preservative in certain foods. Its efficacy and low toxicity is conducive to its use as a preservative and disinfectant. As a preservative, thymol can be incorporated into food products or packaging to inhibit the growth of common foodborne pathogens such as Salmonella spp., thus extending shelf life and enhancing food quality assurance. For work surfaces, thymol is a key active ingredient in botanical disinfectants and sanitizers, providing an alternative to traditional chemical agents. Thymol-based products are effective for decontaminating food-contact and non-food-contact surfaces, reducing the risk of cross-contamination in food processing facilities, kitchens, and other environments.”

4. Please highlight the advantages of flow cytometry over viable plate count method.

Based on the text that appears on lines 348-352, namely “The RAPID-B TPC flow cytometric assay can distinguish between live and dead cells [20, 25]. Most recently, the RAPID-B TPC flow cytometric assay, when compared to traditional growth plates, demonstrated that Salmonella serotypes are capable of surviving in distilled water for over 160 days at both 25°C and 4°C [20]. In this study, Salmonella serotypes treated with thymol failed to grow on TSA plates, but were detected using the RAPID-B TPC assay.” We believe this information sufficiently addresses the concern raised by the reviewer.

5. Language editing: There are some grammar errors (for instance line 180). A thorough grammar and style review is recommended to improve readability (e.g., subject-verb agreement, article usage).

Thank you for your suggestion! To ensure clarity and precision, this manuscript has undergone a meticulous language revision and proofreading as you have recommended it. We intentionally maintained a descriptive writing style, to facilitate the reproducibility of our experimental methods and results by other researchers in the field. Our aim was to provide a comprehensive account of our findings, prioritizing accuracy, and detail.

---

## [Decision Letter · Decision Letter 1]

26 Aug 2025

Examining the viability of five Salmonella enterica subsp. enterica in thymol at 4°C and 25°C using flow cytometry

PONE-D-25-33553R1

Dear Dr. Ahn,

We’re pleased to inform you that your manuscript has been judged scientifically suitable for publication and will be formally accepted for publication once it meets all outstanding technical requirements.

Kind regards,

Jorddy Neves Cruz

Academic Editor

PLOS ONE

Additional Editor Comments (optional):

Reviewers' comments:

Reviewer's Responses to Questions

**Comments to the Author**

1. If the authors have adequately addressed your comments raised in a previous round of review and you feel that this manuscript is now acceptable for publication, you may indicate that here to bypass the “Comments to the Author” section, enter your conflict of interest statement in the “Confidential to Editor” section, and submit your "Accept" recommendation.

Reviewer #1: All comments have been addressed

Reviewer #2: All comments have been addressed

2. Is the manuscript technically sound, and do the data support the conclusions?

Reviewer #1: Yes

Reviewer #2: Yes

3. Has the statistical analysis been performed appropriately and rigorously? 

Reviewer #1: Yes

Reviewer #2: Yes

4. Have the authors made all data underlying the findings in their manuscript fully available?

Reviewer #1: Yes

Reviewer #2: Yes

5. Is the manuscript presented in an intelligible fashion and written in standard English?

Reviewer #1: Yes

Reviewer #2: Yes

6. Review Comments to the Author

Reviewer #1: (No Response)

Reviewer #2: The revisions have been carefully addressed, and the manuscript has been improved accordingly. I have no additional comments.

7. PLOS authors have the option to publish the peer review history of their article (what does this mean? ). If published, this will include your full peer review and any attached files.

**Do you want your identity to be public for this peer review?** For information about this choice, including consent withdrawal, please see our Privacy Policy .

Reviewer #1: **Yes: ** Megan L. Fay

Reviewer #2: No

---

## [Editor Report · Acceptance letter]

PONE-D-25-33553R1

PLOS ONE

Dear Dr. Ahn,

I'm pleased to inform you that your manuscript has been deemed suitable for publication in PLOS ONE. Congratulations! Your manuscript is now being handed over to our production team.

Kind regards,

on behalf of

Dr. Jorddy Neves Cruz

Academic Editor

PLOS ONE